# Clinical Measurements with Calibrated Instance-Dependent Confidence Intervals

**Rotem Nizhar**[1]                                    ROTEMNIZHAR10@GMAIL.COM
[1]*Computer Science Dprt. Tel-Aviv University*

**Lior Frenkel**[2]                                    LIORFRENKEL1992@GMAIL.COM
**Jacob Goldberger**[2]                                JACOB.GOLDBERGER@BIU.AC.IL
[2]*Engineering Faculty, Bar-Ilan University*

**Editors:** Accepted for publication at MIDL 2025

## Abstract

Reporting meaningful confidence intervals for the predictions of a regression neural network is critical in medical imaging applications since clinical decisions rely on network predictions. We expect to obtain larger intervals for difficult examples and smaller ones for easier examples to predict. We demonstrate that training a Gaussian regression network, followed by a non-parametric conformal prediction technique to scale the estimated variance, is the most effective way to achieve a small confidence interval with a coverage guarantee. Through extensive experiments on various medical imaging datasets and network architectures, we show that this combined training and calibration procedure produces improved results compared to previous methods. Our code is publicly available[1].

**Keywords:** confidence interval, uncertainty calibration, conformal prediction

## 1. Introduction

Regression neural networks, which predict continuous quantities, have been a major focus of research in many areas including medical diagnosis (Leibig et al., 2017), weather forecasting (Scher and Messori, 2018) and autonomous driving (Carvalho et al., 2015). Regression neural networks are applied in medical image analysis to measure the size of pathological lesions and the size of anatomical parts and the distance between them. One example is ultrasound-based automatic estimation of fetal biometry which is used to assess the growth and well-being of the fetus (Avisdris et al., 2022). Another example is estimating the bone age of pediatric patients based on radiographs of their hand (Halabi et al., 2019).

The performance of neural network systems has improved dramatically in recent years. However, for safety-critical embodied applications, accurate prediction alone is not sufficient. Uncertainty estimates are important in a wide range of applications, and reporting the confidence of a prediction is essential for reliable and interpretable models. One widely adopted approach to conveying uncertainty is confidence intervals, which enclose the "true value" with a specified probability. The size of these intervals is expected to be small and linked to the case's complexity.

Standard regression networks are trained by minimizing the Mean Squared Error (MSE). These networks provides prediction intervals that have the same length for all test examples, and thus potentially cannot directly report an instance-based confidence interval. However,

---

1. https://github.com/rotem1023/Calibrated-Instance-Dependent

Nizhar[1] Frenkel[2] Goldberger[2]

it is much more informative to provide larger confidence intervals for difficult examples and smaller ones for easier examples to predict. An alternative to the MSE approach is to predict the mean and variance simultaneously. The training loss is then formed by the negative Gaussian log-likelihood. During the testing phase, assuming a Gaussian distribution, the predicted mean and variance values can be translated into an instance-based confidence interval. Other methods that quantify uncertainty in terms of a confidence interval include Bayesian learning (Sheridan, 2012), quantile regression (Koenker and Bassett Jr, 1978) and ensemble-based methods (Gal and Ghahramani, 2016; Lakshminarayanan et al., 2017) that struggle with computational cost by requiring multiple model inferences. Recent studies have compared these methods and no single method has emerged as consistently superior across all evaluation metrics and tasks (see e.g. (Lanini et al., 2024; Kato et al., 2023)).

Consider a regression model that reports confidence intervals with a claimed coverage of $1 - \alpha$. If $1 - \alpha$ of the intervals indeed contain the true value, the model is called calibrated. Deep networks are not well-calibrated and are known to produce unreliable confidence information (Guo et al., 2017). Several recent methods are available for measuring the calibration of a regression network such as the Expected Normalized Calibration Error (ENCE) (Levi et al., 2022) and the Uncertainty Calibration Error (UCE) (Laves et al., 2020; Küppers et al., 2022). Several studies (Laves et al., 2020; Levi et al., 2022; Frenkel and Goldberger, 2023) have proposed a simple calibration method that scales the predictive variance by optimizing a likelihood criterion on a validation set. However, these calibration methods explicitly assume that the conditional density of the correct value given the image is Gaussian, which is not always the case.

Conformal Prediction (CP) (Vovk et al., 2005; Angelopoulos and Bates, 2023) is a general non-parametric calibration method which, given a confidence value, aims to build a confidence interval such that the probability that the correct value is within this set, is indeed the given value. The Conformalized Quantile Regression (CQR) algorithm (Romano et al., 2019) is a calibrated regression method that directly finds a confidence interval without any parametric assumption of the prediction distribution. It consists of a Quantile Regression (QR) (Koenker and Bassett Jr, 1978) followed by a conformalization step. On one hand, CQR is more adaptive to heteroscedasticity and outliers than a Gaussian regression. On the other hand, the pinball loss which is used to train the Quantile regression is less reliable and stable than the Gaussian loss.

CQR is considered the state-of-the-art CP-calibration method to obtain a calibrated instance-based confidence interval of a prediction obtained by a neural network. However, we are not aware of any comparative research on CQR performance either in medical or non-medical data, (see e.g., a discussion in a recent review paper (Kato et al., 2023)). In the current study, we first analyze and demonstrate the advantages and disadvantages of the current methods. We then propose a calibration strategy based on training a Gaussian network followed by a parameter-free CP calibration of the computed variance. We show that this strategy improves the confidence calibration procedure. We report extensive experiments on several medical imaging regression tasks and network architectures that support this combined training and calibration procedure.

## 2. Calibrating Regression Networks

In this section we review parametric and non-parametric methods for instance-dependent calibration of a regression network, which are related to the proposed method. Consider a regression network that outputs mean $\hat{y} = \mu(x)$ and variance $\sigma^2(x)$ for each input image $x$. The mean represents the value predicted by the network, while the variance is the level of uncertainty in the prediction. The network output can be viewed as a Gaussian distribution in the form of $y|x \sim \mathcal{N}(\mu(x), \sigma^2(x))$. Given labeled training data $(x_1, y_1), ..., (x_n, y_n)$, the network is trained by minimizing the loss function:

$$L(\theta) = -\sum_{t=1}^{n} \log \mathcal{N}(y_t; \mu_\theta(x_t), \sigma_\theta^2(x_t)) \tag{1}$$

such that $\theta$ is the network parameter set. From the distribution $y \sim \mathcal{N}(\mu(x), \sigma^2(x))$ we can extract a confidence interval. Define, $\varphi(a) = \int_{-a}^{a} f(z) dz$ such that $z \sim \mathcal{N}(0, 1)$ and $f(z)$ is its density function. For example, a 90% confidence interval for the prediction $\mu(x)$ is defined by:

$$\{y \mid c|y - \mu(x)|/\sigma(x) \le \varphi^{-1}(0.9)\}. \tag{2}$$

Since the variance $\sigma^2(x)$ is predicted by the neural network, it may not be well-calibrated and may be either underestimated or overestimated. The training loss function (1) assumes Gaussian distribution, which may not be correct so that the conversion from variance to confidence interval (2) can be wrong. Gaussian Variance Scaling (Gaussian-VS) (Laves et al., 2020; Levi et al., 2022; Frenkel and Goldberger, 2023) is a method to calibrate the variance $\sigma(x)$ that yields a meaningful confidence interval. This method is based on a Gaussian assumption of the conditional density $f(y|x)$. Gaussian-VS computes a scalar $r$, which scales the variance predicted by the network: $\sigma(x) \to r \cdot \sigma(x)$. Given labeled validation data $(x_1, y_1), ..., (x_n, y_n)$, we look for a scalar $r$ that minimizes the loss:

$$L(r) = -\sum_{t=1}^{n} \log \mathcal{N}(y_t; \mu_\theta(x_t), r^2 \sigma_\theta^2(x_t)). \tag{3}$$

It is easy to verify that the optimal $r$ is:

$$\hat{r}^2 = \frac{1}{n} \sum_{t=1}^{n} s_t^2 \qquad \text{s.t.,} \qquad s_t = \frac{|y_t - \mu(x_t)|}{\sigma(x_t)}. \tag{4}$$

Given a test image $x$, the calibrated confidence interval with coverage $1 - \alpha$ is:

$$[\mu(x) - \varphi^{-1}(1 - \alpha/2) \cdot \hat{r} \cdot \sigma(x), \mu(x) + \varphi^{-1}(1 - \alpha/2) \cdot \hat{r} \cdot \sigma(x)]. \tag{5}$$

Neural networks tend to be very robust to model mismatch and inaccurate ground truth measurements due to their high non-linearity and over-parameterization. In contrast, Gaussian-VS calibration is linear and consists of a single parameter. Thus if $f(y|x)$ is uni-modal but not Gaussian, while network training works well, Gaussian-VS fails to produce an accurate confidence interval.

NIZHAR[1] FRENKEL[2] GOLDBERGER[2]

The CQR calibration approach (Romano et al., 2019) consists of a non-parametric Quantile Regression (QR) (Koenker and Bassett Jr, 1978) followed by a conformalization step. Define the $\gamma$-quantile (pinball) loss:

$$L_\gamma(y, t) = 1_{\{t<y\}}(y - t)\gamma + 1_{\{t>y\}}(t - y)(1 - \gamma).$$

Given a training set $(x_1, y_1), \ldots, (x_n, y_n)$ the QR algorithm trains a $\gamma$-quantile estimation $\hat{t}_\gamma(x)$ using the pinball loss:

$$L_\gamma(\theta) = \frac{1}{n} \sum_{i=1}^{n} L_\gamma(y_i, \hat{t}_\gamma(x_i, \theta)),$$

such that $\theta$ is the network parameter set. Given a miscoverage rate $\alpha$, we train a QR network with two heads $\hat{t}_{\alpha/2}(x)$ and $\hat{t}_{1-\alpha/2}(x)$ to obtain an instance-dependent confidence interval: $[\hat{t}_{\alpha/2}(x), \hat{t}_{1-\alpha/2}(x)]$. The CQR algorithm applies a CP procedure to ensure that the coverage is indeed $1 - \alpha$. Define the following conformal score:

$$s(x, y) = \max\{\hat{t}_{\alpha/2}(x) - y, y - \hat{t}_{1-\alpha/2}(x)\}.$$

Let $s_1, \ldots, s_n$ be the conformal scores of a given validation set $(x_1, y_1), \ldots, (x_n, y_n)$ and let $q$ be the $(1 - \alpha)$ quantile of $s_1, \ldots, s_n$. The calibrated confidence interval is:

$$C_q(x) = [\hat{t}_{\alpha/2}(x) - q, \hat{t}_{1-\alpha/2}(x) + q].$$

Unlike Gaussian-VS, the interval obtained by CQR has a coverage guarantee. The CP theory (Vovk et al., 2005) guarantees that: $1 - \alpha \leq p(y \in C_q(x)) \leq 1 - \alpha + \frac{1}{n-1}$ where $y$ is the (unknown) true label. Note that this is a marginal probability over all possible test points and coverage may be worse or better for some cases. It can be proved that conditional coverage is, in general, impossible (Foygel Barber et al., 2021). QR is much more difficult to train than a Gaussian regression network and when QR produces poor interval estimations, the performance of CQR is also affected since it tries to cover the guaranteed validity by sacrificing efficiency (Chung et al., 2021; Kato et al., 2023).

## 3. CP-based Variance Scaling

In this section we present a method that combines the benefits of the two methods described above. We first train a parametric Gaussian network to predict the target and its variance and then apply a non-parametric CP to calibrate the estimated variance. Assume we trained a regression network that outputs mean $\hat{y} = \mu(x)$ and variance $\sigma^2(x)$ for each input image $x$ as described in Section 2. Given a threshold $1 - \alpha$, we can apply CP to scale the variance and find a confidence interval around $\mu(x)$ in the form of:

$$C_q(x) = [\mu(x) - q \cdot \sigma(x), \mu(x) + q \cdot \sigma(x)] \tag{6}$$

such that the true value $y$ is within this interval with probability $1 - \alpha$, i.e., $p(y \in C_q(x)) = 1 - \alpha$. The calibration parameter $q$ is found in the following way. For each labeled data $(x, y)$ define the conformal score: $s = |y - \mu(x)|/\sigma(x)$. It is easy to verify that:

$$C_s(x) = [\mu(x) - |y - \mu(x)|, \mu(x) + |y - \mu(x)|]$$

---

**Algorithm 1** Conformal-Prediction based Variance Scaling (CP-VS)

---

    **input:** A labeled dataset divided into training and validation subsets and a confidence level $1-\alpha$.

    - Train a regression network $x \to (\mu_\theta(x_t), \sigma_\theta^2(x))$ by minimizing the loss

$$L(\theta) = -\sum_{t=1}^{n} \log \mathcal{N}(y_t; \mu_\theta(x_t), \sigma_\theta^2(x_t))$$

    - Compute the conformal scores on the validation set:

$$s_t = |y_t - \mu(x_t)|/\sigma(x_t), \qquad t = 1, ..., n$$

    - Sort the scores $s_1 \le s_2, ..., \le s_n$ and set $q = s_{\lceil (n+1)(1-\alpha) \rceil}$.

    - The confidence interval of a new test point $x$ is: $C(x) = [\mu(x) - q \cdot \sigma(x), \mu(x) + q \cdot \sigma(x)]$.

    - There is a marginal coverage guarantee: $p(y \in C(x)) \ge 1 - \alpha$.

---

and $y \in C_q(x)$ if and only if $q \ge s$. In other words, $C_s(x)$ is the minimal interval centered at $\mu(x)$ which contains the true value $y$. Let $s_1, ...., s_n$ be the conformal scores of the validation set $(x_1, y_1), ..., (x_n, y_n)$ respectively. The calibration value $q$ computed by the CP algorithm is the $\frac{\lceil (n+1)(1-\alpha) \rceil}{n}$ quantile of $s_1, ..., s_n$. In other words, $q$ is the minimal value for which the true value lies within the confidence interval defined by $q$ in the $(1 - \alpha)$ portion of the validation set. The CP theory (Vovk et al., 2005) guarantees that regardless of the data distribution, for test data $(x, y)$, the value $q$ found by the CP algorithm satisfies:

$$1 - \alpha \le p(y \in C_q(x)) \le 1 - \alpha + \frac{1}{n-1} \tag{7}$$

such that $n$ is the size of the validation set.

    Note that both Gaussian-VS and CP-VS calibrate by scaling the estimated standard deviation $\sigma(x)$ using a scalar value that is learned from the same conformal scores $s_1, ..., s_n$ (4) obtained from the validation set. Gaussian-VS is based on the scores' average while CP-VS is based on a quantile of the scores. In case the conditional density of the target value given the input image is indeed Gaussian, the two calibration methods asymptotically coincide. In case the conditional density is not Gaussian, the quantile is a more effective measure than the mean when constructing a confidence interval. The CP-VS algorithm is summarized in Algorithm Box 1.

    The CP-VS has several benefits. Firstly, network training is conducted using the robust Gaussian loss (unlike CQR which uses the pinball loss). Secondly, CP-VS achieves calibration through the CP procedure which has a parameter-free theoretical coverage guarantee (unlike Gaussian-VS which has neither a theoretical nor practical coverage guarantee).

## 4. Experimental Results

In this section, we empirically compare the performance of the confidence intervals computed by CP-VS to those computed by Gaussian-VS and CQR in terms of both interval length and coverage.

NIZHAR[1] FRENKEL[2] GOLDBERGER[2]

Table 1: Calibration results measured by average confidence interval length and coverage (%). The method that reports the minimal length among those who have a coverage guarantee is shown in bold.

| $1-\alpha = 0.9$ | | Gaussian-VS | | CQR | | CP-VS | |
|---|---|---|---|---|---|---|---|
| Dataset | Architecture | length ↓ | coverage | length ↓ | coverage | length ↓ | coverage |
| BoneAge | DenseNet-201 | $0.184 \pm 0.003$ | $91.10 \pm 0.71$ | $0.411 \pm 0.005$ | $90.02 \pm 0.78$ | $\mathbf{0.176} \pm 0.003$ | $89.92 \pm 0.85$ |
| | EfficientNet-B4 | $0.190 \pm 0.003$ | $90.19 \pm 0.56$ | $0.602 \pm 0.007$ | $90.02 \pm 0.83$ | $\mathbf{0.189} \pm 0.003$ | $89.95 \pm 0.72$ |
| OCT | DenseNet-201 | $0.128 \pm 0.001$ | $99.94 \pm 0.06$ | $0.106 \pm 0.002$ | $90.11 \pm 0.99$ | $\mathbf{0.048} \pm 0.001$ | $90.47 \pm 1.32$ |
| | EfficientNet-B4 | $0.127 \pm 0.001$ | $99.93 \pm 0.06$ | $0.157 \pm 0.003$ | $89.97 \pm 1.50$ | $\mathbf{0.050} \pm 0.001$ | $90.07 \pm 1.80$ |
| Brain | DenseNet-201 | $0.316 \pm 0.002$ | $83.30 \pm 0.54$ | $0.581 \pm 0.002$ | $89.80 \pm 0.43$ | $\mathbf{0.371} \pm 0.004$ | $90.01 \pm 0.56$ |
| | EfficientNet-B4 | $0.411 \pm 0.002$ | $90.84 \pm 0.34$ | $0.631 \pm 0.007$ | $90.01 \pm 0.47$ | $\mathbf{0.396} \pm 0.004$ | $90.01 \pm 0.54$ |
| DLS1 | DenseNet201 | $0.102 \pm 0.004$ | $93.55 \pm 1.32$ | $0.278 \pm 0.003$ | $89.68 \pm 1.01$ | $\mathbf{0.092} \pm 0.001$ | $90.33 \pm 1.04$ |
| | EfficientNet-B4 | $0.086 \pm 0.008$ | $96.11 \pm 2.21$ | $0.350 \pm 0.004$ | $90.37 \pm 1.16$ | $\mathbf{0.068} \pm 0.001$ | $89.87 \pm 0.98$ |
| DLS2 | DenseNet201 | $0.063 \pm 0.001$ | $91.25 \pm 0.72$ | $0.316 \pm 0.002$ | $90.36 \pm 0.89$ | $\mathbf{0.060} \pm 0.001$ | $89.79 \pm 0.87$ |
| | EfficientNet-B4 | $0.071 \pm 0.001$ | $92.56 \pm 0.68$ | $0.299 \pm 0.002$ | $89.93 \pm 0.78$ | $\mathbf{0.064} \pm 0.001$ | $89.94 \pm 1.13$ |
| DLS3 | DenseNet201 | $0.157 \pm 0.004$ | $91.53 \pm 1.02$ | $0.178 \pm 0.002$ | $90.18 \pm 0.90$ | $\mathbf{0.151} \pm 0.003$ | $90.18 \pm 0.88$ |
| | EfficientNet-B4 | $0.076 \pm 0.005$ | $94.41 \pm 1.85$ | $0.299 \pm 0.002$ | $89.96 \pm 0.78$ | $\mathbf{0.065} \pm 0.001$ | $90.12 \pm 1.06$ |
| DLS4 | DenseNet201 | $0.093 \pm 0.006$ | $93.30 \pm 1.70$ | $0.220 \pm 0.002$ | $89.92 \pm 0.99$ | $\mathbf{0.082} \pm 0.001$ | $89.86 \pm 1.08$ |
| | EfficientNet-B4 | $0.077 \pm 0.004$ | $94.07 \pm 1.74$ | $0.328 \pm 0.005$ | $90.21 \pm 0.93$ | $\mathbf{0.069} \pm 0.001$ | $90.22 \pm 0.84$ |
| DLS5 | DenseNet201 | $0.115 \pm 0.001$ | $90.45 \pm 0.73$ | $0.227 \pm 0.003$ | $90.19 \pm 0.89$ | $\mathbf{0.114} \pm 0.002$ | $89.99 \pm 1.09$ |
| | EfficientNet-B4 | $0.066 \pm 0.002$ | $92.96 \pm 0.84$ | $0.474 \pm 0.011$ | $89.75 \pm 1.14$ | $\mathbf{0.058} \pm 0.002$ | $89.70 \pm 1.27$ |

| $1-\alpha = 0.95$ | | Gaussian-VS | | CQR | | CP-VS | |
|---|---|---|---|---|---|---|---|
| Dataset | Architecture | length ↓ | coverage | length ↓ | coverage | length ↓ | coverage |
| BoneAge | DenseNet-201 | $0.219 \pm 0.004$ | $95.40 \pm 0.43$ | $0.539 \pm 0.011$ | $94.97 \pm 0.87$ | $\mathbf{0.224} \pm 0.004$ | $94.91 \pm 0.53$ |
| | EfficientNet-B4 | $0.227 \pm 0.003$ | $94.71 \pm 0.43$ | $0.489 \pm 0.008$ | $95.08 \pm 0.58$ | $\mathbf{0.233} \pm 0.004$ | $95.26 \pm 0.47$ |
| OCT | DenseNet-201 | $0.153 \pm 0.001$ | $99.94 \pm 0.06$ | $0.146 \pm 0.005$ | $95.07 \pm 1.20$ | $\mathbf{0.057} \pm 0.001$ | $94.96 \pm 0.87$ |
| | EfficientNet-B4 | $0.152 \pm 0.002$ | $99.98 \pm 0.02$ | $0.235 \pm 0.006$ | $95.12 \pm 0.69$ | $\mathbf{0.059} \pm 0.001$ | $94.99 \pm 1.06$ |
| Brain | DenseNet-201 | $0.377 \pm 0.001$ | $89.78 \pm 0.30$ | $0.786 \pm 0.000$ | $95.68 \pm 0.16$ | $\mathbf{0.518} \pm 0.009$ | $95.04 \pm 0.44$ |
| | EfficientNet-B4 | $0.380 \pm 0.002$ | $91.15 \pm 0.34$ | $0.754 \pm 0.008$ | $94.97 \pm 0.41$ | $\mathbf{0.498} \pm 0.010$ | $94.96 \pm 0.38$ |
| DLS1 | DenseNet201 | $0.121 \pm 0.005$ | $96.97 \pm 0.73$ | $0.362 \pm 0.004$ | $95.09 \pm 0.67$ | $\mathbf{0.110} \pm 0.002$ | $95.35 \pm 0.67$ |
| | EfficientNet-B4 | $0.102 \pm 0.010$ | $98.38 \pm 1.10$ | $0.425 \pm 0.004$ | $95.20 \pm 0.65$ | $\mathbf{0.080} \pm 0.001$ | $94.87 \pm 0.63$ |
| DLS2 | DenseNet201 | $0.075 \pm 0.001$ | $96.02 \pm 0.44$ | $0.352 \pm 0.003$ | $95.14 \pm 0.71$ | $\mathbf{0.071} \pm 0.001$ | $94.81 \pm 0.56$ |
| | EfficientNet-B4 | $0.084 \pm 0.001$ | $95.52 \pm 0.43$ | $0.379 \pm 0.003$ | $94.88 \pm 0.58$ | $\mathbf{0.081} \pm 0.002$ | $94.92 \pm 0.61$ |
| DLS3 | DenseNet201 | $0.183 \pm 0.005$ | $95.74 \pm 0.63$ | $0.172 \pm 0.003$ | $95.17 \pm 0.78$ | $\mathbf{0.179} \pm 0.003$ | $95.24 \pm 0.59$ |
| | EfficientNet-B4 | $0.090 \pm 0.007$ | $97.74 \pm 1.13$ | $0.432 \pm 0.006$ | $95.06 \pm 0.47$ | $\mathbf{0.078} \pm 0.001$ | $95.15 \pm 0.63$ |
| DLS4 | DenseNet201 | $0.111 \pm 0.007$ | $96.70 \pm 1.02$ | $0.251 \pm 0.003$ | $95.05 \pm 0.72$ | $\mathbf{0.100} \pm 0.002$ | $94.96 \pm 0.82$ |
| | EfficientNet-B4 | $0.092 \pm 0.005$ | $97.73 \pm 0.84$ | $0.363 \pm 0.008$ | $94.97 \pm 0.63$ | $\mathbf{0.080} \pm 0.001$ | $95.02 \pm 0.73$ |
| DLS5 | DenseNet201 | $0.138 \pm 0.001$ | $95.33 \pm 0.46$ | $0.283 \pm 0.003$ | $95.21 \pm 0.59$ | $\mathbf{0.134} \pm 0.002$ | $94.76 \pm 0.65$ |
| | EfficientNet-B4 | $0.079 \pm 0.002$ | $96.16 \pm 0.54$ | $0.552 \pm 0.012$ | $94.70 \pm 0.64$ | $\mathbf{0.073} \pm 0.002$ | $94.77 \pm 0.68$ |

**Datasets.** We implemented the proposed calibration methods on several medical imaging regression tasks to evaluate their performance. The experimental setup follows the one used in (Laves et al., 2020) and includes the following medical datasets:

- BoneAge - Hand CT age regression from the RSNA pediatric bone age dataset (Halabi et al., 2019). The task here is to infer a person's age in months from CT scans of the hand. This dataset is the largest used in this study and has 12,811 images, from which we used 6811/2000/4000 images for training/validation/testing.

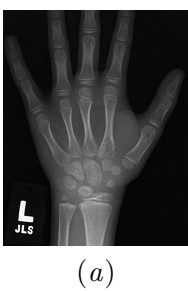 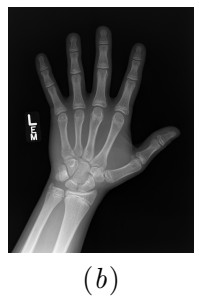 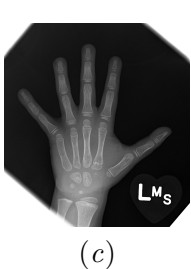

(a)           (b)           (c)

Figure 1: Samples from the BoneAge test set. (a) Target is 0.47, intervals are - Gaussian-VS: [0.37,0.57], CP-VS: [0.38,0.56], CQR: [0.20,0.83]. (b) Target is 0.76, intervals are - Gaussian-VS: [0.67,0.79], CP-VS: [0.68,0.78], CQR: [0.23,0.81]. (c) Target is 0.21, intervals are - Gaussian-VS: [0.17,0.31], CP-VS: [0.17,0.31], CQR: [0.24,0.90].

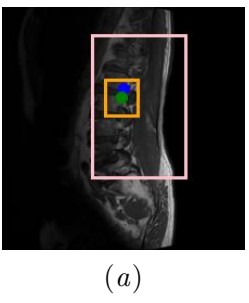 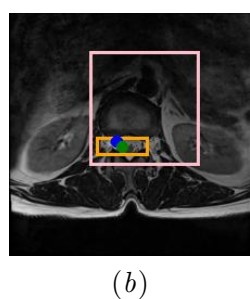 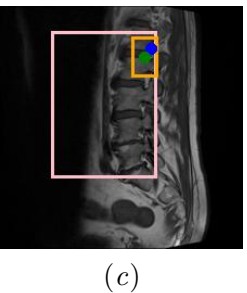

(a)           (b)           (c)

Figure 2: Samples from the DLS-1 test set. Blue - true position of the lumbar, green - predicted position of the lumbar, orange - bounding box created by CP-VS, pink - bounding box created by CQR.

- OCT - Six degrees of freedom (6DoF) needle pose estimation on optical coherence tomography (OCT). This dataset contains 5,000 3D-OCT scans with the accompanying needle pose $y \in [0, 1]^6$, from which we use 3300/850/850 for training/validation/testing (Laves et al., 2020).

- Brain - We used the brain tumor dataset from the Medical Segmentation Decathlon (Simpson et al., 2019; Antonelli et al., 2022), which consists of 484 brain MRI scans with corresponding tumor segmentation masks. The dataset was split into training, validation, and test sets in an 80%/20%/20% ratio. Each scan is a 3D volume of size $240 \times 240 \times 155$. We extracted individual slices from each MRI scan, resulting in 155 image slices of size $240 \times 240$ per scan. A regression target was assigned to each image by counting the number of labeled brain tumor pixels (Gustafsson et al., 2023).

- DLS - This dataset is designed to facilitate the detection and classification of degenerative lumbar spine (DLS) conditions using MRI images. Each image includes annotations for the $(x, y)$ positions of five vertebrae. For each vertebra, the dataset is

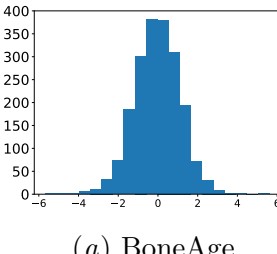
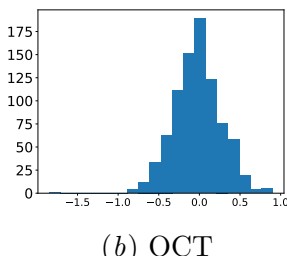
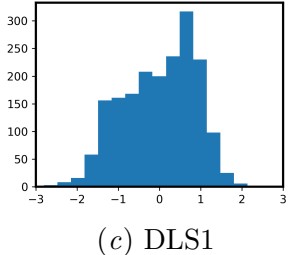

$(a)$ BoneAge $\qquad$ $(b)$ OCT $\qquad$ $(c)$ DLS1

Figure 3: Histograms of the normalized network prediction values computed on the validation sets using DenseNet-201.

divided into 60% for training, 20% for validation, and 20% for testing, with approximately 10,000 images per vertebra (Richards et al., 2024).

**Implementation details.** The network architectures used were EfficientNet-B4 (Tan and Le, 2019) and DenseNet-201 (He et al., 2016; Guo et al., 2017). The last linear layer of all networks was replaced by two linear layers predicting the mean and log-variance. The networks were trained until no further decrease of the loss on the validation set could be observed. During training, each input $x$ was passed through the network 25 times, with a dropout applied, resulting in variations among the outputs. The final prediction used in the loss function was computed as the average of these 25 outputs. To implement the CQR algorithm, we used the code from the CQR project GitHub[2]. CQR is trained to minimize the average length while keeping the coverage valid. Note that a different QR network must be trained for each value of the threshold $1 - \alpha$. In contrast, in the case of CP-VS, the Gaussian network is trained only once, and only the CP step needs to be redone for each threshold.

**Evaluation measures.** The standard direct way to evaluate the performance of confidence-interval estimators on a given test set $(x_1, y_1), ..., (x_n, y_n)$ is by computing the degree of coverage and the average length of the prediction intervals. Smaller average interval widths indicate higher precision. The length and coverage are formally defined as:

$$\text{length} = \frac{1}{n} \sum_i |C(x_i)|, \quad \text{coverage} = \frac{1}{n} \sum_i \mathbf{1}(y_i \in C(x_i))$$

such that $C(x_i)$ is the confidence interval of $x_i$ and $|C(x_i)|$ is its length. The best algorithm is the one that reports the minimal average interval length among those that satisfy the coverage requirement.

**Results.** Table 1 shows the comparative calibration results (length and coverage) for the three methods (Gaussian-VS, CQR and CP-VS) on the test set. The results were averaged over 20 random splits of the data into validation and test sets. Both CP-VS and CQR, which apply a CP procedure, obtained the exact required coverage, as guaranteed by the CP theorem. However, the average length reported by the CQR was much larger due to the non-robust training of the QR algorithm. The coverage rate of Gaussian-VS was

---

2. https://github.com/yromano/cqr

inconsistent. In some cases, it was below the required coverage $(1 - \alpha)$; in other cases, it was above (resulting in a large average interval). Note that Gaussian-VS has no theoretical coverage guarantee.

**Visual examples.** We next illustrate the proposed method on several examples. Figure 1 shows examples from the BoneAge dataset. The figures illustrate the same trend that was reported in Table 1, namely that the CP-VS yields confidence intervals with the smallest size. Next we illustrate results of predicting the positions of lumbar L1/L2 using the DLS-1 dataset. Two networks were trained, one for each dimension, and the CP procedure was applied to guarantee 95% coverage for each dimension. As a result, 90% of the bounding boxes produced by CP-VS and CQR will accurately encompass the true position of the lumbar. Fig. 2 presents examples of bounding boxes around lumbar position predictions computed by CP-VS and CQR on images from the test set. Notably, CP-VS produces considerably smaller bounding boxes.

**Normality check.** We next analyzed whether the output distribution of the Gaussian network was indeed Gaussian for each dataset. For each image $x$ in the validation set, we computed the scalar $(y - \mu(x))/\sigma(x)$ such that $y$ was the correct value and $\mu(x)$ and $\sigma(x)$ were predicted by the network. Note that $y \sim \mathcal{N}(\mu(x), \sigma^2(x))$ implies that $(y - \mu(x))/\sigma(x) \sim \mathcal{N}(0, 1)$. The histograms of the three datasets (BoneAge, OCT and DLS) are shown in Fig. 1. We also applied the Kolmogorov–Smirnov test to check the data's normality and obtained 0.018 (BoneAge), 1.42e-60 (OCT), and 0.001 (DLS1). Hence, the BoneAge task histogram was the only one that resembled a Normal distribution and passed the Gaussianity test. We can see in Table 1 that when the normal assumption is valid, Gaussian-VS works well and produces an effective confidence interval. However, when the normal assumption fails Gaussian-VS has inconsistent behaviors. In some cases it doesn't satisfy the coverage requirement and in other cases it yields large confidence intervals.

## 5. Conclusions

In this study, we addressed the problem of reporting a reliable confidence interval that varies in size across the images and reflects instance-specific uncertainty with a theoretical coverage guarantee that makes it useful in real systems. We proposed a CP-based procedure that calibrates the prediction of a Gaussian network. We showed that CP-VS produces a confidence interval whose average is much smaller than CQR while maintaining the same coverage guarantee. We focused here on medical imaging applications, but the conclusions are general and relevant for calibrating any regression network. CP-based Calibration algorithms (in both classification and regression setups) are not robust to real-world situations of missing data, label noise (Einbinder et al., 2024) and distribution shift (Gustafsson et al., 2023). Possible future research directions include extending the proposed method in a way that allows it to handle these problems.

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
