# OpenReview forum: "Clinical Measurements with Calibrated Instance-Dependent Confidence Interval"
_MIDL.io/2025/Conference — MIDL 2025 Poster_

### Official Review · Reviewer_e5Cn · 2025-02-16

**Confidence:** 3
**Preliminary Rating:** 3

**Summary:**

In this paper, the authors propose CP-VS, which is a method to calibrate confidence interval in medical imaging regression tasks. The method combines Gaussian network training with conformal prediction (CP) to get calibrated uncertainty estimates. The authors further evaluate CP-VS on various medical datasets including bone age estimation, needle post tracking, and spine measurements, where they found that CP-VS is capable of generating tighter confidence intervals while sustaining required coverage probability.

**Strengths:**

First, the paper addresses an important problem in medical imaging applications where uncertainty needs to be reliably quantified for clinical deployment. Hence, the technical approach is well-motivated and builds upon established methods, such as Gaussian network and conformal predictions. The existing literature was discussed, which lays good theoretical groundwork for CP-VS.

In addition, the presentation is clear and well-structured for easy follow-through. The math formulation is precise and adequately explained and documented for easy reproduction. The experiment results were also clearly documented in Table 1.

The authors also provide ample empirical evidence for CP-VS backed by experiment results. What's commendable is that the result evaluation goes beyond metric reporting, but also includes investigation of Gaussian assumption through normality tests as well as reproducible  implementation details.

**Weaknesses:**

The literature review and experiment comparison seem to focus on Gaussian VS and CQR, whereas more recent uncertainty quantification methods including Bayesian neural network and probabilistic deep ensembles.

The paper could benefit from more discussion on the practicality of CP-VS given the importance of runtime considerations in clinical deployment. Recognized that the CP-VS shows tighter intervals - how does the computational overhead compare with other approach?

In addition, regarding data selection, the datasets used are relatively controlled. While this is not necessarily a weakness, it'd be helpful to discuss if authors foresee challenges in deploying CP-VS to more challenging real-world clinical data. How would noise and missing info affect CP-VS results?

**Detailed Comments:**

See details in strengths and weaknesses sections.

**Justification Of The Preliminary Rating:**

CP-VS is a novel concept. However, it is unclear why the authors didn't include more recent uncertainty justification methods in literature review or experiment. Depending on the timeline, revision/re-submission might not be possible.

**Questions To Address In The Rebuttal:**

• How would CP-VS compare to other uncertainty quantification methods such as Bayesian NN and probabilistic deep ensembles?
	• How does CP-VS perform with significant noise and missing value?
	• What is the runtime consideration for CP-VS (vs. existing approach)?
	• Why does CP-VS work well even when Gaussian assumption is violated?

---

> ### Author Response · Authors · 2025-03-06
>
> We thank the reviewer for the comments.
>
> Regarding Bayesian neural network and probabilistic deep ensembles. Unlike CP, these methods are heuristic, with no theoretical guarantee regarding their true value coverage. Hence, it is problematic to compare them with CP-based methods. We propose a CP-based method that works consistently and significantly better than CQR.
>
> We added runtime considerations. Although all CP-based methods offer the desired coverage guarantee, CQR uses a difficult-to-train loss function (pinball loss), resulting in large CP intervals. Furthermore, a different QR network must be trained for each threshold value.
>
> Handling noise and missing values are indeed natural directions for future research. There are several recent studies on  CP with noisy labels in classification setups dealing with these issues.  In regression tasks, it is still an open problem.  We added a comment on this in the conclusion section.
>
> CP-VS works well even when the Gaussian assumption is violated. Network training is less sensitive to model mismatch due to the highly nonlinear nature of the network. However, a calibration method that explicitly assumes a Gaussian distribution may produce poor results if this assumption is incorrect. The key advantage of CP is that it is distribution-free.

---

> ### Comment · Area_Chair_u6UD · 2025-03-17
> **Final rating is missing**
>
> Dear e5Cn,
>
> Would you be so kind to provide your final rating? Thanks!
>
> - AC

---

### Official Review · Reviewer_vw4s · 2025-02-21

**Confidence:** 4
**Preliminary Rating:** 3
**Recommendation:** Poster
**Final Rating:** 4

**Summary:**

The authors study how to train regression models with calibrated uncertainty estimates.

They propose "_Conformal-Prediction based Variance Scaling (CP-VS)_", which is basically a combination of "_Gaussian Variance Scaling (Gaussian-VS)_" and "_Conformalized Quantile Regression (CQR)_". They first train a Gaussian model that outputs $\mu(x)$ and $\sigma(x)$, and then a scaling factor $q$ is fit using a conformalization step on validation data, producing the final prediction interval $C(x)$.

They compare their proposed CP-VS with Gaussian-VS and CQR in terms of interval coverage and average interval length, using three image-based regression datasets with different types of medical images.

Their CP-VS generally achieves the smallest intervals among methods which reach the desired interval coverage.

**Strengths:**

- The studied problem is interesting and relevant, uncertainty estimation and reliability for _regression_ models is also understudied compared to classification.

- The paper is well written overall and quite easy to follow, things are clearly described.

- The experimental evaluation is quite solid with reasonable baseline methods, three relatively large image-based regression datasets with different types of medical images, and two different network architectures.

- The proposed CP-VS method is conceptually quite simple, I think it makes sense to apply a conformalization step to Gaussian-VS.

- The proposed CP-VS performs well compared to the baselines.

**Weaknesses:**

- The authors write that "_The CP theory (Vovk et al., 2005) guarantees that regardless of the data distribution, for test data (x, y), the value q found by the CP algorithm satisfies..._" in Section 3. However, this marginal coverage guarantee is only valid under the assumption of exchangeably drawn train/val and test data (true for i.i.d. data, for instance). I think this is a quite crucial condition, that should be made more clear in the paper, also in Algorithm 1.

- Related to the above point, it seems as if all datasets utilized in the experimental evaluation are split i.i.d into train/val/test, such that the marginal coverage guarantee indeed should hold. However, as demonstrated e.g. in [_How Reliable is Your Regression Model's Uncertainty Under Real-World Distribution Shifts?_  (TMLR, 2023)](https://openreview.net/forum?id=WJt2Pc3qtI), there are many real-world examples where this does not hold, and in these cases also conformal-based methods can produce highly overconfident intervals (clearly failing to satisfy the desired coverage).

- Overall, the proposed method and evaluation is similar to what is studied in "_How Reliable is Your Regression Model's Uncertainty Under Real-World Distribution Shifts?_", but the authors don't cite this paper. The claim "_CQR is considered the state-of-the-art method to obtain a calibrated instance-based confidence interval of a prediction obtained by a neural network. However, we are not aware of any comparative research, either in medical or non-medical data, (see e.g., a discussion in a recent review paper (Kato et al., 2023))_" in Section 1 might therefore not be entirely accurate.

**Detailed Comments:**

Questions/suggestions:
- In "_How Reliable is Your Regression Model's Uncertainty Under Real-World Distribution Shifts?_" they evaluate a method which is similar to the proposed CP-VS (conformalization of a Gaussian model, see eq. (4) and (7)), it would be interesting to compare these two?

- It would be interesting to extend your evaluation with one of the medical datasets from "_How Reliable is Your Regression Model's Uncertainty Under Real-World Distribution Shifts?_", to see how the evaluated methods perform in the distribution shift setting?

- If you need space to add more discussions/results, Section 2 could probably be shortened a bit (e.g., by using more inline equations)?




Minor things:
- "1. introduction", typo.

- "2. Calibrating regression networks" --> "2. Calibrating Regression Networks"?

- Figure 1, "(c) DLS1" --> "(c) DLS"? Or at least describe briefly somewhere what is meant by DLS1, ..., DLS5?

- Section 2, "Gaussian-Vs computes a scalar", Gaussian-Vs --> Gaussian-VS?

- Section 2, "Given a miss-coverage rate", miss-coverage --> miscoverage?

- Section 4, "was reported in Tables 1 and 2" --> "was reported in Table 1"? Same also in "We can see in Table 1 and Table 2 that when" later.

**Justification Of The Final Rating:**

I quite like the paper overall, and the studied problem is interesting and important. Uncertainty estimation and reliability for _regression_ models is understudied compared to classification, I think it deserves more attention from the MIDL audience.

**Justification Of The Preliminary Rating:**

I quite like this paper overall, I think the studied problem is interesting and important. I want to be able to recommend accept.

However, I think the current version requires some clarifications, more discussion of related work etc.

**Questions To Address In The Rebuttal:**

See Weaknesses and Questions/suggestions above.

At the very least, the assumptions for the marginal coverage guarantee should be clarified (under what conditions the proposed CP-VS can be expected to perform well), and the relation to "_How Reliable is Your Regression Model's Uncertainty Under Real-World Distribution Shifts?_" should be discussed a bit somewhere in the paper.

**Special Issue:**

No

---

> ### Author Response · Authors · 2025-03-06
>
> Thank you for the comments.
>
> Domain shift forms a major challenge for many deep learning methods. As the paper [1] demonstrates, domain shift causes uncertainty reporting in the regression model to be less calibrated. This is also true for the CP-VS algorithm. Handling domain-shift problems in regression models is beyond the scope of our study and is a challenging future research direction. We added a comment on this issue in the conclusion section.
>
> We note that our method can also be used to calibrate the ensemble method ([1}, Eq. 5) by replacing the Gaussian-based calibration ([1] Eq 4) with a CP-based calibration.
>
> Our focus was on instance-based confidence intervals with a theoretical coverage guarantee. As far as we know, CP is currently the only way to obtain such a guarantee. In [1]  they evaluated a simple CP method where the confidence interval is fixed. In our study, we extended it to an image-based interval by applying CP on the estimated variance.
>
> [1] How reliable is your regression...., Gustafsson et al., Trans on machine learning research, 2023.

---

> > ### Comment · Reviewer_vw4s · 2025-03-12
> >
> > Thank you for the response.
> >
> > I have read the other reviews and all responses.
> >
> > I don't think the other reviews give my any obvious reasons to change my score.
> >
> > The author responses are not overly comprehensive/detailed, but the updated paper contains results on another dataset etc.
> >
> > The rebuttal has not fully addressed my concerns:
> >
> > "The authors write that _"The CP theory (Vovk et al., 2005) guarantees that regardless of the data distribution, for test data (x, y), the value q found by the CP algorithm satisfies..."_ in Section 3. However, this marginal coverage guarantee is only valid under the assumption of exchangeably drawn train/val and test data (true for i.i.d. data, for instance). I think this is a quite crucial condition, that should be made more clear in the paper, also in Algorithm 1."
> > - - I still think this should be clarified in Algorithm 1.
> >
> > I still don't think that the claim _"CQR is considered the state-of-the-art method to obtain a calibrated instance-based confidence interval of a prediction obtained by a neural network. However, we are not aware of any comparative research, either in medical or non-medical data, (see e.g., a discussion in a recent review paper (Kato et al., 2023))"_ in Section 1 is entirely accurate.
> >
> > If these two points are addressed, I will increase my score to 4: Weak accept.

---

> > > ### Author Response · Authors · 2025-03-12
> > >
> > > Thank you for the comments.
> > >
> > > In the final version, we will clarify that the marginal coverage guarantee of the proposed algorithm is valid only under the assumption of exchangeability.
> > >
> > > We will also remove the sentence: "CQR is considered the state-of-the-art method."
> > >
> > > We are not aware of any recent comparison between calibration methods for constructing confidence intervals in regression prediction.

---

> > > > ### Comment · Reviewer_vw4s · 2025-03-14
> > > >
> > > > Thank you for the response.
> > > >
> > > > I will increase my score to 4: Weak accept.

---

### Official Review · Reviewer_Sd2x · 2025-02-22

**Confidence:** 5
**Preliminary Rating:** 4
**Recommendation:** Poster
**Final Rating:** 4

**Summary:**

Effective uncertainty quantification in clinical settings should yield larger confidence intervals for challenging predictions and smaller ones for simpler cases. The paper reveals that Gaussian-VS is unsuitable for medical use, as its confidence intervals lack statistical reliability. It also finds that CQR produces suboptimal results, with overly wide average intervals. In contrast, the authors show that CP-VS—integrating Gaussian loss training with distribution-free conformal prediction calibration—outperforms both, delivering the most accurate and efficient confidence intervals.

**Strengths:**

1 The paper is highly clinically relevant: It addresses a critical need in medical imaging by developing methods for reliable uncertainty quantification, directly applicable to clinical decision-making in areas like bone age assessment and spinal condition detection.
2 The paper conducts a rigorous comparative study: It systematically evaluates its proposed CP-VS method against established techniques (Gaussian-VS and CQR), using comprehensive experiments across diverse datasets and network architectures to ensure robust and credible findings.
3 The findings are compelling and insightful: The results highlight CP-VS’s ability to produce smaller, statistically meaningful confidence intervals with guaranteed coverage, offering a significant advancement over prior methods and sparking interest for further exploration in regression network calibration.

**Weaknesses:**

1 Enhance the visual appeal of figures. It may be too much to ask, but The paper could improve the aesthetic quality of its figures (e.g., histograms in Figure 1, samples in Figure 2)

2 Validate findings across additional datasets in future works.

3 Clarify code availability is unclear.

**Detailed Comments:**

This paper demonstrates strong clinical relevance, offering valuable insights for medical imaging applications. The findings are both intriguing and practically useful, highlighting advancements in uncertainty quantification. For future work, the authors are encouraged to enhance the visual quality of figures and validate their conclusions by testing the method on additional datasets and clinical scenarios. Additionally, making the code publicly available would further strengthen the study’s impact and reproducibility.

**Justification Of The Final Rating:**

The paper offers a fresh take on the clinical applicability of uncertainty and is highly relevant. It might be preliminary, but it could be the start of something great.
I will keep my original rating.

**Justification Of The Preliminary Rating:**

1 The paper is highly clinically relevant: It addresses a critical need in medical imaging by developing methods for reliable uncertainty quantification, directly applicable to clinical decision-making in areas like bone age assessment and spinal condition detection.
2 The paper conducts a rigorous comparative study: It systematically evaluates its proposed CP-VS method against established techniques (Gaussian-VS and CQR), using comprehensive experiments across diverse datasets and network architectures to ensure robust and credible findings.
3 The findings are compelling and insightful: The results highlight CP-VS’s ability to produce smaller, statistically meaningful confidence intervals with guaranteed coverage, offering a significant advancement over prior methods and sparking interest for further exploration in regression network calibration.

**Questions To Address In The Rebuttal:**

1 Can the same findings and conclusions be extended to other clinical applications?
2 Will the code be available during the conference?: Is the source code for Gaussian-VS and CP-VS, as mentioned in the paper, accessible to attendees at the MIDL 2025 conference

**Special Issue:**

No

---

> ### Author Response · Authors · 2025-03-03
>
> Thank you for your comments.
>
> In the final version, we'll improve the visual quality of the figures.
> The project code will be available as part of the final version.

---

### Official Review · Reviewer_A9t6 · 2025-02-24

**Confidence:** 3
**Preliminary Rating:** 4
**Recommendation:** Poster
**Final Rating:** 4

**Summary:**

For image-based regression problems (like extracting biomarkers or other quantities like age or even coordinates), calibration translates into fulfilling confidence intervals given certain coverage: if a regression model (like predicting mean +/- variance in direct aleatoric uncertainty estimation nets) predicts narrower intervals than those they later produce, then it is miscalibrated. The authors use a conformal prediction layer over one such system to calibrate it, improving upon previous similar techniques on several datasets.

**Strengths:**

- Conformal Prediction for regression is not so immediate and well-explored as for classification.
- As with all CP methods, you do not need to modify the underlying model and training details in order to profit from it.
- Evaluation takes into account joint coverage and confidence interval width, instead of blindly looking at coverage which is trivial to get right by just increasing width. Instance-dependence is also a good feature of the method.
- Thanks for making the code available (maybe mention in the abstract?)

**Weaknesses:**

- I cannot find super-serious weak points in this work. If any, I do not agree with the sentence "The network architectures used where EfficientNet B4 (2019) and DenseNet-201 (2016), which are state of the art for deep models" - these are good convnets, but they are no longer the best choice in 2025. Some research has shown that transformers have different calibration properties than CNNs, so it would have been great to have an experiment using them, instead of two cnns.

**Detailed Comments:**

- Not a problem of the authors, but this sort of regression calibration is quite niche I think. Is calibration critical when regressing (X,Y) coordinates on vertebra localization problem? Or on needle pose estimation on OCT? I mean, if a method is under/overconfident in these problems (where non-clinical quantities like age or other biomarkers), how useful is this? If I get a wide interval on a coordinate, is this useful info for a practitioner, like they should not trust the localization?
- Maybe clarify why sometimes there is more than one bold-faced method in Table 1? I'm guessing the second best is not statistically different than the best one, but it's not mentioned anywhere I think.

**Justification Of The Final Rating:**

To be honest, I find the authors' two-line response quite underwhelming. I suggested to remove DenseNet/EfficientNet and instead add a Transfomer-based model (because it is 2025 and their calibration properties have been shown to be different from ConvNets'), but they just answer that they will add a Transformer model in the final version, but why not in the updated version they uploaded, since they had time to run another experiment with a different datasetr? I also asked for some extra clinical motivation of why we need calibration in non-biomarker image-based regression (coordinates, etc.), but no effort has been given to this. Even the simple fix of explaining why there are several methods boldfaced in the main table has not been addressed.

I have also looked into the other reviewer's rating, their comments and the authors' answers. I'll keep my WA score, but I must say have been tempted to downgrading my rating in view of the very cursory author's reaction to my and others' comments.

**Justification Of The Preliminary Rating:**

I don't think the paper has very relevant weakness. The topic is a bit super-specialized, so some extra justification of the clinical need of avoiding miscalibration in tasks related to localization would be welcome.

**Questions To Address In The Rebuttal:**

What I said above, I guess.

**Special Issue:**

No

---

> ### Author Response · Authors · 2025-03-03
>
> Thank you for your comments.
>
> In the final version, we will add the results of calibrating a transformer-based regression network in addition to CNN. We note that even if an architecture can yield a more calibrated network, there
> is still a need for a systematic calibration procedure.
> .

---

### Author Rebuttal · Authors · 2025-03-06

**Rebuttal:**

We uploaded a revised version of the paper that addressed the reviewers' comments. The changes are marked in red.
1) We added a link to the project code in the abstract.
2) We added a review of previous methods for reporting reliable confidence interval.
3) We added another experiment on a brain images.
4) We added another visual illustration of the results of the CP-VS.
5) We added a discussion on possible future research directions.

**Supporting Material:**

/attachment/f4e8f62bb40222138e720c0a85c06b87b0ab30b3.pdf

---

### Comment · Area_Chair_u6UD · 2025-03-08
**Time for discussion and review of the rebuttal**

Dear reviewers,

It is now time to consider the responses from the authors. If you are or are not satisfied with author's reply please still post to openreview your feedback to the rebuttal and update your scores. Especially, please update the scores if you feel that the authors have addressed your concerns.

Please note that you can and **are encouraged** to discuss the scores of other reviewers if you disagree with them to make the best

As AC, my responsibility is to post meta-reviews by March 21st, and I would thus like to kindly ask you to consider the authors' rebuttal as soon as possible.

// Your Area Chair

---

### Meta-Review · Area_Chair_u6UD · 2025-03-24

**Recommendation:** Accept (Poster)
**Confidence:** 3

**Metareview:**

Most 3/4 reviewers recommend weak accept, and reviewer e5Cn did not take part in the final rating / updating the score.

I looked at the paper, and it looks at an important problem uncertainty calibration in regression networks. While I personally find the rebuttal quite underwhelming, I think that the paper has merit if the authors will really add the transformer experiments they have promised.

/ Area Chair